# The Bone Microenvironment Soil in Prostate Cancer Metastasis: An miRNA Approach

**DOI:** 10.3390/cancers15164027

**Published:** 2023-08-09

**Authors:** Anne Natalie Prigol, Michele Patrícia Rode, Fernanda da Luz Efe, Najla Adel Saleh, Tânia Beatriz Creczynski-Pasa

**Affiliations:** Department of Pharmaceutical Sciences, Federal University of Santa Catarina, Florianopolis 88040-900, Santa Catarina State, Brazil; anne.prigol@posgrad.ufsc.br (A.N.P.); michele.rode@ufsc.br (M.P.R.); fernanda.efe@posgrad.ufsc.br (F.d.L.E.); adelsale@msu.edu (N.A.S.)

**Keywords:** bone metastasis, bone microenvironment, prostate cancer, miRNAs, exosomes

## Abstract

**Simple Summary:**

Prostate cancer (PCa) is the second most incident cancer in men worldwide. Despite having high cure rates when locally confined, PCa has a high risk of mortality in advanced stages, owing to the few treatment options for the metastatic disease, which occurs mostly in bones. Tumor progression seems to be related to deregulation of microRNA (miRNA) expression. These small noncoding RNA molecules act as posttranscriptional regulators of gene expression in donor cells or distant sites (by exosome transportation), preparing the future metastatic niche. Identification of suitable miRNAs may assist in an early and less invasive diagnosis and drug therapy, positively impacting patient quality of life and improving our understanding of the molecular aspects of bone metastasis.

**Abstract:**

Bone metastatic prostate cancer (PCa) is associated with a high risk of mortality. Changes in the expression pattern of miRNAs seem to be related to early aspects of prostate cancer, as well as its establishment and proliferation, including the necessary steps for metastasis. Here we compiled, for the first time, the important roles of miRNAs in the development, diagnosis, and treatment of bone metastasis, focusing on recent in vivo and in vitro studies. PCa exosomes are proven to promote metastasis-related events, such as osteoblast and osteoclast differentiation and proliferation. Aberrant miRNA expression in PCa may induce abnormal bone remodeling and support tumor development. Furthermore, miRNAs are capable of binding to multiple mRNA targets, a dynamic property that can be harnessed for the development of treatment tools, such as antagomiRs and miRNA mimics, which have emerged as promising candidates in PCa treatment. Finally, miRNAs may serve as noninvasive biomarkers, as they can be detected in tissue and bodily fluids, are highly stable, and show differential expression between nonmetastatic PCa and bone metastatic samples. Taken together, the findings underscore the importance of miRNA expression profiles and miRNA-based tools as rational technologies to increase the quality of life and longevity of patients.

## 1. Introduction

Prostate cancer (PCa) is the second most incident cancer in men worldwide, with an estimated 1.4 million new cases and 375,000 deaths in 2020 [1]. Despite recent advances in PCa treatment, metastasis is a common problem during the course of the disease, and the leading cause of death. Because PCa is initially silent, many patients diagnosed with PCa already have metastatic sites [2,3], typically in the lymph nodes, liver, lungs, adrenal glands, and bones [4,5]. The bones are the most common site of metastasis. Metastatic PCa cells were shown to exhibit tropism for the bone microenvironment, explained by the seed and soil hypothesis postulated by Steven Paget in 1889 [6]. PCa bone metastasis is a critical complication associated with severe bone lesions, pathological fractures, spinal cord compression, and hyperkalemia, leading to severe pain and untreatable consequences [7,8]. Patients with PCa bone metastasis have a significantly higher mortality risk, with a 5-year survival rate of only 28% [9].

Interactions between tumor cells and the bone microenvironment are crucial for the establishment and growth of PCa cells. Osteoclasts and osteoblasts, cells responsible for bone homeostasis and remodeling [10], modulate the bone microenvironment and thus determine the metastatic phenotype [8,11]. Research has shown that microRNAs (miRNAs) play an important role in this process via the regulation of osteoblast and osteoclast activities and differentiation [12]. miRNAs are small noncoding RNAs that regulate posttranscriptional gene expression through mRNA complementarity, which is fundamental to PCa progression, metastasis development, and phenotypic expression [12,13]. Moreover, miRNAs serve as an important intercellular communication tool via exosomes, a subtype of extracellular vesicles (EVs) secreted by several cell types, including cancer cells. Exosomes were found to contribute to bone metastasis by promoting cell migration, invasion, and remodeling and establishing a premetastatic niche [14,15]. Exosomes derived from PCa cells carry DNA, mRNA, noncoding RNA (ncRNA), miRNA, lipids, and proteins and transfer these constituents to the bone microenvironment, creating a fertile site for cancer cell growth [14,16,17].

In this review, we aimed to describe the role of the bone microenvironment in PCa metastasis, examine how PCa-derived miRNAs and exosomes can contribute to creating a cancer-favorable microenvironment in the premetastatic niche and regulating the metastatic PCa phenotype, and investigate the role of miRNAs in the treatment, prognosis, and diagnosis of PCa bone metastasis.

## 2. Bone Metastasis Microenvironment

The bone microenvironment is a dynamic compartment that undergoes remodeling throughout life. Bone remodeling can be understood as a coordinated process of bone resorption and formation resulting from the action of two types of bone cells, namely osteoclasts and osteoblasts [18]. Osteoclasts degrade the bone matrix in response to receptor activator of nuclear factor kappa-Β ligand (RANKL) and macrophage colony-stimulating factor (M-CSF) signaling [19]. Osteoblasts are responsible for bone formation [20]. These cells produce extracellular matrix (ECM) proteins such as type I collagen, osteocalcin, and alkaline phosphatase. A subset of osteoblasts differentiates into osteocytes. Osteocytes control bone remodeling by modulating interactions between osteoblasts and osteoclasts in response to hormonal, morphogenic, and mechanical signals. During the process of bone remodeling, osteocytes may express RANKL, leading to bone resorption via osteoclast activation, or may decrease the expression of Dickkopf-related protein 1 (DKK1) and sclerostin, which promotes increased bone formation through osteoblast activation via Wingless-related integration site (Wnt) signaling [21,22,23].

Bone formation and resorption are critical for bone health, and an imbalance in the control of these processes, such as increased resorption or suppression of bone formation, is associated with bone metastasis. Bone is the most frequent site of metastasis in prostate and breast cancers [24,25]. Evidence indicates that PCa cells may exploit certain aspects of the bone microenvironment for homing by fostering the formation of premetastatic niches, even from a distance [26].

Tumor cells may home to the bone during the development of the primary tumor, remaining latent until metastatic relapse occurs. Several factors of the bone microenvironment regulate the reactivation of tumor cells in the bone. In bone metastasis, paracrine crosstalk between cancer and bone cells constitutes a vicious osteolytic cycle [27,28,29,30]. Tumor cells concurrently suppress osteoblasts and induce massive bone destruction by osteoclasts. Aberrant expression of RANKL and M-CSF by cancer cells has been shown to induce osteoclast activation, leading to bone degradation and the release of growth factors (e.g., IGF-1 and TGF-β) in the bone matrix, stimulating tumor growth [30,31,32].

## 3. Exosomes and the Future Bone Metastatic Niche

Exosomes are a small lipid bilayer subtype of EVs (30 to 150 nm in diameter). Secreted by eukaryotic cells after the fusion of multivesicular bodies with the plasma membrane, exosomes may contain DNA, mRNA, ncRNA, miRNA, lipids, and proteins [17,33]. Initially, exosomes were believed to be responsible for carrying cellular waste out of cells [34]. It was only in 1996 [35] that these vesicles began to be seen as important actors in cellular communication. Exosomes are currently understood to be fundamental components for communication between tumor cells and the tumor microenvironment. Many cells secrete miRNAs via exosomes, modulating activities in recipient cells through the horizontal transfer of information [16]. The exact function of exosomes depends on the type of donor cell.

In the tumor context, cell-derived exosomes have been shown to target and prepare the future metastatic niche by transferring oncogenic proteins and genes to nontumor cells [15,36,37]. In the premetastatic niche, exosomes can upregulate inflammatory molecules [38], remodel the architecture of the ECM [39], increase angiogenesis, and promote vascular permeability [40,41,42,43]. Moreover, tumor-derived exosomes can determine metastatic organotropism. The targeting of a specific organ depends primarily on the uptake of tumor exosomes by organ cells. Local uptake may be mediated by differential integrin expression, as observed by Hoshino et al. [37], who demonstrated that α6β4 and α6β1 exosomal integrins mediate lung metastasis, whereas αvβ5 is associated with liver metastasis and αvβ6 regulates matrix metalloproteinase-2 (MMP2), promoting osteolytic processes in PCa bone metastasis [44].

Tumoral exosomes seem to have a strong affinity for the bone microenvironment. Systematic delivery of murine melanoma B16510 exosomes results in localization to bones and “educates” bone marrow progenitor cells, increasing metastatic behavior [15]. Exosomes derived from a highly metastatic breast cancer cell line increased osteoclast activity and reduced bone density, accelerating bone lesions in order to reconstruct the microenvironment for bone metastasis [45]. Multiple myeloma-derived exosomes were found to contain large amounts of amphiregulin (AREG) and induce osteoclastogenesis [46]. Osteoclast differentiation mediated by AREG contained in non-small cell lung cancer exosomes has also been observed [47].

In the PCa context, exosomes can mediate osteogenic or osteolytic metastasis through mechanisms not yet fully elucidated [48]. Pyruvate kinase M2 (PKM2) expression is associated with clinical metastasis and may be an important inducer of premetastatic niche formation, evidenced by the fact that PKM2 transfer from PCa exosomes to bone marrow stromal cells regulates stromal cell-derived factor 1 (SDF-1) and directs the bone marrow to contribute to premetastatic niche formation [49]. Probert et al. [50] reported that osteoblasts grown in co-culture with PCa cells benefit from tumor-derived EVs, which increase osteoblast viability and produce a more supportive growth environment. Conditioned medium from the PCa androgen-independent PC-3 cell line induces the expression of the osteoclastogenesis-associated genes insulin-like growth factor-binding protein 5 (IGFBP-5), interleukin 6 (IL-6), monocyte chemoattractant protein 1 (MCP-1), and RANKL, promoting osteoclastogenesis in RAW 264.7 cells [51]. PC-3 EVs also contain Ets1, which is an osteoblast differentiation-related transcription factor that is transferred to osteoblasts when they are cultured with tumoral microvesicles, inducing osteoblast differentiation [52]. Exosomes from murine PCa cells decrease the fusion and differentiation of monocytic osteoclast precursors to mature, multinucleated osteoclasts and also decrease osteoclast fusion and proliferation markers, such as dendritic cell-specific transmembrane protein (DCSTAMP), triiodothyronine receptor auxiliary protein (TRAP), cathepsin K, and MMP9 [53].

The success of exosomal communication between cells depends on vesicle internalization. Inder et al. [54] showed that the presence of caveolae-associated protein 1 (CAVIN-1) in EVs derived from PC-3 cells reduces vesicle internalization efficiency in the osteoclast precursor cells RAW 264.7 compared to control PC3 EVs, resulting in failure to induce osteoblast proliferation. Interestingly, bioinformatics analysis of the proteome of EVs derived from nonmineralizing and mineralizing osteoblasts demonstrated that these vesicles act in pathways related to cell survival and growth, which was confirmed in vitro by increased proliferation of PC-3 cells after osteoblast EV uptake [55], suggesting a feedback cycle between PCa and the tumor bone interface. These findings pave the way for further research to prevent tumor spread and improve the treatment of bone metastasis.

## 4. miRNAs and Bone Remodeling

Bone remodeling is a complex process regulated by numerous biological factors and signals responsible for bone structure renewal, whereby old bone tissue is continually replaced by new bone tissue. Remodeling results from a balance between bone formation (stimulated by osteoblasts) and bone resorption (stimulated by osteoclasts) [56,57], and disruption of this balance can lead to bone disease [10]. Bone formation and bone resorption are interconnected: cells of the osteoblast lineage activate the bone remodeling cycle by releasing enzymes and factors on the bone’s surface that modulate osteoclast activation and differentiation, prompting these cells to begin the resorption process. As bone resorption occurs, recruited osteoclasts emit signals for osteoblast differentiation and migration [56,58,59].

Once PCa disseminates to the bone and cancer cells start to proliferate at the metastatic site, normal bone cell function is altered, and the balance between bone formation and bone resorption is disturbed. The phenotype of bone metastasis can be determined according to the balance between osteoblast and osteoclast activities, classified as osteolytic, osteoblastic, or mixed lesions [60]. Metastasis of PCa cells is most often osteoblastic, as it stimulates osteoblast activities by disrupting the bone microenvironment, supporting bone formation [61]. However, patients characterized by an osteoblastic phenotype of PCa bone metastasis can also exhibit osteolytic lesions [62]. This is explained by the fact that, for successful bone colonization, PCa cells need to modulate both osteoblastic and osteolytic processes. Thus, PCa cells produce a variety of factors that can act directly or indirectly on osteoblast and osteoclast activity to support bone metastasis development [63,64,65,66]. These findings suggest that bone colonization by PCa necessitates a general increase in bone remodeling.

miRNAs are responsible for regulating several pathways related to bone remodeling by directly or indirectly controlling cell signaling for osteoclast and osteoblast activity and differentiation [67,68,69]. Therefore, aberrant miRNA expression in PCa can lead to abnormal bone remodeling and stimulate tumor development and bone metastasis [12]. The role of miRNAs in bone remodeling during PCa bone metastasis is not fully understood, and their molecular targets are still being elucidated. However, some studies have demonstrated the importance and role of miRNAs in regulating osteoblast and osteoclast differentiation and activity during PCa bone metastasis, influencing bone remodeling and metastatic phenotype [12,70]. A summary of the currently known roles of miRNAs in bone remodeling during PCa bone metastasis and in the crosstalk between cancer cells and the bone microenvironment is illustrated in Figure 1.

### 4.1. miRNAs Related to Osteoblast Activity in PCa

The osteoblast plays a central role in bone formation by producing several constituents of the bone matrix and differentiating into osteocytes, a crucial cell for bone remodeling [56,71]. Osteoblasts originate from the differentiation of multipotent mesenchymal stem cells (MSCs), which are able to differentiate into multiple cell lineages, such as adipocytes, chondrocytes, myocytes, and fibroblasts. This process is regulated by numerous transcription factors [72,73].

One of the most important regulators of osteoblast differentiation is runt-related transcription factor 2 (RUNX2), a protein that regulates the differentiation of MSCs into immature osteoblasts and modulates bone matrix protein expression during osteoblast differentiation [74,75]. Deregulation of RUNX2 seems to be important for osteolytic and osteoblastic metastasis [76], and some miRNAs were found to alter RUNX2 expression in PCa bone metastasis. miR-466 expression, for instance, is downregulated in PCa tissues. This miRNA seems to act as a suppressor of PCa proliferation and bone metastasis through RUNX2 regulation. Overexpression of miR-466 in PCa cells impairs migration and invasive capacity via RUNX2 inhibition, inducing the downregulation of several RUNX2 target genes related to migration and bone metastasis [77]. miR-203 has a similar role in PCa bone metastasis. Saini et al. [78] demonstrated that miR-203 expression is downregulated in bone metastatic PCa cells and that its upregulation attenuates bone metastasis via negative regulation of RUNX2. Ectopic expression of miR-203 seems to downregulate osteocalcin and osteopontin genes (osteoblastic genes related to the maintenance of bone maturation, mineralization, and bone remodeling [79]) and inhibit the expression of Distal-less homeobox 5 (DLX5, a protein responsible for RUNX2 activation [80]). Moreover, miR-203 was demonstrated to inhibit the expression of suppressor of mothers against decapentaplegic homolog 4 (SMAD4), an important regulator of TGF-β signaling. These observations suggest that miR-203 and miR-466 downregulation enhances the expression of RUNX2 and its regulatory genes in bone metastatic PCa cells, promoting bone formation and metastasis in PCa.

In cancer, osteoblasts can secrete chemokines and growth factors that play an important role in attracting PCa cells to bone tissues, promoting metastasis [61]. Tai et al. [81] hypothesized that osteoblast-derived factors can downregulate miR-126 and induce migration of human PCa cells. Wnt1-induced secreted protein 1 (WISP-1), an important regulator of bone development and repair [82], is highly secreted by osteoblasts and can regulate αvβ1 integrin, focal adhesion kinase (FAK), and p38 signaling pathways, leading to inhibition of miR-126 expression in PCa cells. miR-126 negatively regulates vascular cell adhesion molecule 1 (VCAM-1), an adhesion molecule that modulates the motility of human PCa cells [81]. With miR-126 downregulation via osteoblast-derived WISP-1, VCAM-1 expression is upregulated, increasing the ability of PCa cells to migrate to the bone.

TGF-β was demonstrated to have an important role in the development and progression of PCa bone metastasis. Siu et al. [83] observed that miR-96 expression was enhanced by TGF-β signaling via SMAD-dependent transcription in PCa bone metastasis. Furthermore, miR-96 directly targeted and downregulated AKT1S1, leading to increased mechanistic targeting of rapamycin kinase (MTOR) activity. MTOR is involved in bone homeostasis and development through MTORC1 and MTORC2, which stimulate osteoblast differentiation and function [84]. Additionally, as demonstrated by Voss et al. [85], miR-96 enhances PCa cell–cell interactions and their ability to bind to osteoblasts by upregulating E-cadherin and epithelial cell adhesion molecule (EPCAM) expression. miR-96 is overexpressed in PCa bone metastasis [83,85] and seems to promote bone metastasis in PCa tissues by increasing osteoblast activity and differentiation.

The expression of Wnt ligands was found to be upregulated in an osteoblastic metastatic PCa cell line, inducing the activation of the canonical Wnt pathway and increasing bone formation [86]. Wnt pathways regulate several processes that modulate osteogenesis and influence osteoblastic bone metastasis in PCa [61,86,87]. Some miRNAs seem to modulate Wnt signaling during PCa progression. For instance, Li et al. [88] demonstrated that miR-218 directly targets leucine-rich repeat-containing G-protein-coupled receptor 4 (LGR4), an IL-6 responsive gene associated with cancer progression, and modulates phosphoinositide 3-kinase (PI3K)/protein kinase B (AKT) and Wnt/β-catenin pathways in LNCaP cells treated with exogenous IL-6 (LNCaP-IL-6^+^). miR-218 is underexpressed in PCa and seems to act as a tumor suppressor [88,89,90]. Therefore, the ability of miR-218 to prevent the increase in cell proliferation and invasion in LNCaP-IL-6^+^ cells promoted by IL-6 is believed to stem from inhibition of Wnt/β-catenin pathways.

Chen et al. [91] evidenced that miR-34a expression is suppressed by Ras signaling in PCa cells. Ras signaling is upregulated in metastatic samples and seems to be an important pathway in PCa metastasis and epithelial–mesenchymal transition (EMT) [92]. Downregulation of miR-34a leads to cell growth and invasion, given that miR-34a inhibits the expression of transcription factor 7 (TCF7), a Wnt/β-catenin pathway activator [93,94]. miR-34a may be related to inhibition of osteoblast differentiation and activity through Wnt signaling inhibition and TCF7 regulation in PCa cells.

Exosomal miRNAs derived from PCa cells could promote osteogenic differentiation of human bone mesenchymal stem cells (hBMSC) [95]. One possible mechanism for this process was proposed by Mo et al. [95], who showed that the expression of long noncoding RNA nuclear-enriched abundant transcript 1 (lncRNA-NEAT1) is enhanced in hBMSC by PCa-derived exosomes and contributes to osteogenic differentiation of hBMSC via miR-205-5p regulation. miR-205-5p negatively regulates RUNX2 expression, suppressing osteogenic differentiation in hBMSC and inhibiting bone formation [96]. miR-205-5p expression is downregulated in PCa bone metastasis [95,97] and seems to be related to lncRNA-NEAT1, which acts as a competing endogenous RNA of this miRNA, promoting RUNX2 expression and contributing to osteogenic differentiation. Moreover, Hashimoto et al. [70] reported that miR-940 is highly expressed in exosomes derived from the C4-2B PCa cell line and promotes osteogenic differentiation of human mesenchymal stem cells (hMSC) in vitro by targeting Rho GTPase-activating protein 1 (ARHGAP1) and reticulophagy regulator family member 2 (FAM134A). ARHGAP1 negatively regulates hMSC activity via suppression of the RhoA/ROCK pathway, which is responsible for stimulating osteogenic differentiation in hMSC [98] and EMT [99]. FAM134A, also known as MAG-2, is associated with the promotion of metastatic ability in lung cancer cells [100]. Nevertheless, the osteogenesis-related functions and mechanisms of FAM134A remain unclear.

Yu et al. [101] demonstrated that exosomes and exosomal miRNAs derived from osteoblastic, osteoclastic, and mixed PCa cell lines can induce bone lesions and modulate PCa bone disease progression. Analysis of exosomes isolated from PCa cell lines revealed miR-92a-1-5p as the most abundant miRNA. miR-92a-1-5p has several key roles in bone homeostasis. It induces the degradation of type 1 collagen by negatively regulating COL1A1, inducing bone ECM degradation, inhibiting osteoblastogenesis, promoting osteoclast differentiation, and enhancing bone resorption in PCa cells. Conversely, it was observed that miR-148a-3p and miR-375, also highly expressed in PCa-derived exosomes, enhance osteogenic differentiation and have osteoblastic effects [101]. Therefore, these results evidence that exosomes derived from PCa cells can carry both osteoblastic and osteoclastic miRNAs, demonstrating the potential of exosomes in regulating bone metabolism and homeostasis. Accordingly, Li et al. [102] found that miR-375 is upregulated in LNCaP-derived exosomes and suggested that these EVs could enter osteoblasts and enhance miR-375 levels, stimulating osteoblast activity and differentiation. The direct target of miR-375 in PCa tissues has not yet been identified, but it is known that miR-375 overexpression increases osteoprotegerin, RUNX2, osteopontin, and bone sialoprotein gene expression in LNCaP cells, modulating genes that are directly related to osteoblast differentiation and activity [102,103] (Table 1).

### 4.2. miRNAs Related to Osteoclast Activity in PCa

Osteoclasts are responsible for bone resorption. The RANKL/RANK pathway seems to be the most important mechanism related to osteoclast differentiation, being essential for bone metabolism under normal and pathological conditions, including PCa bone metastasis [58,66,104]. RANKL activates RANK, which stimulates osteoclast differentiation, proliferation, and survival through distinct pathways [105]. Nuclear factor-kappa B (NF-κB) is one of the targets activated by RANK and a critical transcription factor in osteoclastogenesis [106]. Activation of NF-κB signaling in PCa cells contributes to bone metastasis via regulation of osteoclastogenic genes by miRNAs [66].

Ren et al. [107] demonstrated that miR-210-3p is overexpressed in bone metastatic PCa tissues and promotes sustained activation of the NF-κB signaling pathway by inhibiting TNFAIP3 interacting protein 1 (TNIP1) and suppressor of cytokine signaling 1 (SOCS1) expression, negative regulators of NF-κB signaling. miR-532-3p is downregulated in metastatic PCa tissues and was shown to act as a bone metastasis suppressor by inhibiting NF-κB activation through TNF receptor-associated factor 1 (TRAF1), TRAF2, and TRAF3 [108]. Likewise, miR-204-5p was demonstrated to act as a tumor suppressor in PCa bone metastasis by inhibiting NF-κB signaling via direct targeting of TRAF1, TGF-β activated kinase 1 binding protein 3 (TAB3), and mitogen-activated protein kinase 3 (MAP3K3) [109], which are responsible for maintaining sustained NF-κB activation [110,111]. Upregulation of miR-204-5p decreases the osteolytic area of metastatic tumors and bone metastatic score [109]. Duan et al. [112] showed that PC-3-derived exosomes can downregulate miR-214 *in vitro*, causing inhibition of osteoclast differentiation by suppressing the NF-κB signaling pathway. However, the targets and regulation mechanisms of miR-214 have not yet been clarified. The hypothesis is that PCa cells can promote bone metastasis by inhibiting osteoclast differentiation and promoting osteoblast differentiation and bone formation. Therefore, miR-210-3p, miR-532-3p, miR-204-5p, and miR-214 might modulate osteoclastogenesis in PCa tissues via NF-κB regulation.

Chen et al. [113] found evidence that thrombospondin-2 (TSP-2) might regulate osteoclastogenesis and bone remodeling by downregulating miR-376c. TSP-2 is a glycoprotein that modulates cell adhesion and migration [114,115] and positively regulates MMP2 expression through downregulation of miR-376c in PCa cells. MMP2 is a direct target of miR-376c and enhances matrix degradation and osteolytic bone metastasis in PCa. TSP-2 downregulates miR-376c expression via mitogen-activated protein kinase 1 (MAPK) pathways, enhancing MMP2 expression, PCa migration, and osteolytic metastasis in vivo [113]. Additionally, oncostatin M (OSM), a member of the IL-6 subfamily, was shown to be a potent regulator of bone remodeling [116]. OSM can promote osteoblast and osteoclast differentiation and is responsible for inducing IL-6 secretion by osteoblasts [116,117]. Han et al. [118] evidenced that the miR-181b-5p/OSM axis regulates osteoclast differentiation and modulates PCa cell proliferation, migration, and invasion. OSM is a direct target of miR-181b-5p, and overexpression of miR-181b-5p downregulates OSM expression, decreasing the production of osteoclastogenic factors such as IL-6 and AREG and increasing that of the anti-osteoclastogenic factor osteoprotegerin, thereby suppressing osteoclast differentiation. OSM overexpression, however, reversed the effects of miR-181b-5p and recovered the expression of osteoclastogenic factors. Therefore, miR-181b-5p can modulate osteoclastic differentiation via regulation of osteoclastogenic factors. On the other hand, miR-133a-3p was demonstrated to reduce osteolytic bone lesions in PCa in vivo via downregulation of the PI3K/AKT signaling pathway, directly targeting multiple cytokine receptors, such as epidermal growth factor receptor (EGFR), fibroblast growth factor receptor 1 (FGFR1), insulin-like growth factor1 receptor (IGF1R), and MET receptor tyrosine kinase [119]. The PI3K/AKT pathway was shown to have an important role in promoting bone metastasis and osteolytic bone lesions in PCa [120]. Therefore, it is probable that miR-133a-3p acts as a suppressor of osteolytic bone metastasis in PCa.

In PCa, TGF-β has a dual function. In the early stages of PCa, TGF-β acts as a tumor suppressor by inhibiting tumor cell proliferation. In advanced stages of PCa, however, TGF-β signaling is dysfunctional, and the cytokine begins to act as a tumor promoter, contributing to PCa metastasis [121,122,123]. TGF is highly expressed in the bone matrix, and TGF-β receptors are expressed in both osteoclasts and osteoblasts [124]. Osteoblasts can secrete latent TGF-β in the bone environment. During osteoclastic bone resorption, latent TGF-β is activated and released from the bone matrix [125,126]. TGF-β recruits MSCs to bone resorption sites through the SMAD signaling pathway, in which MSCs differentiate into osteoblasts [127]. Therefore, TGF-β acts as a coupling factor of bone resorption to bone formation and is an important regulator of osteoblast and osteoclast differentiation via SMAD signaling [128,129]. In osteoclasts, TGF-β activates the SMAD2/3 complex via TGF-β receptors, which bind directly to TRAF6, a downstream mediator of RANK/RANKL, and induces osteoclast differentiation via activation of nuclear factor of activated T-cells (NFATc1) [130]. The SMAD2/3 complex can also bind to SMAD4, activating the transcription of several genes related to osteoclast differentiation [131].

Some miRNAs were reported to suppress osteolytic lesions in PCa tissues by downregulating TGF-β activity via direct modulation of SMAD signaling pathways. Huang et al. [132] demonstrated that overexpression of miR-582-3p and miR-582-5p reduces the bone metastatic osteolytic area via downregulation of SMAD2, SMAD4, TGF-β type I receptor (TGFβRI), and TGF-β type II receptor (TGFβRII) in PCa tissues. miR-505-3p inhibits TGF-β signaling by directly targeting SMAD2 and SMAD3, reducing PCa cell invasion and bone metastasis [133]. miR-19a-3p inhibits invasion, migration, and osteolytic bone lesions in PCa tissues by downregulating SMAD2 and SMAD4, reducing TGF-β signaling activity [134]. miR-133b reduces the osteolytic area of PCa cells in vivo by suppressing TGF-β activity via direct targeting of TGFβRI and TGRFβRII [135]. It was evidenced by Dai et al. [49] that miR-33a-5p is repressed by ZEB1 via activation of TGF-β signaling in PCa. Upregulation of miR-33a-5p in vivo suppressed bone osteolytic lesions and bone metastatic sites in PCa via inhibition of TGF-β signaling, achieved by direct targeting of TGFβRI. Thus, miR-582, miR-505-3p, miR-19a-3p, miR-133b, and miR-33a-5p can modulate osteoclastic differentiation by suppressing the TGF-β signaling pathway (Table 2).

### 4.3. miRNAs Related to Both Types of Bone Metastasis

miRNAs can regulate many target mRNAs, and one mRNA can be regulated by several miRNAs [136]. In PCa cells, some miRNAs were demonstrated to play a role in both osteoclastogenesis and osteoblastogenesis.

miR-100 seems to have both types of action during PCa progression. miR-100 expression is significantly decreased in PCa bone metastasis compared with primary PCa [137,138,139,140]. Wang et al. [141] demonstrated that miR-100 is responsible for inhibiting the expression of Argonaute 2 (AGO2) in PCa tissues, suppressing migration, invasion, and EMT. AGO2 is a key regulator of miRNA biogenesis and is responsible for mediating gene silencing by binding mature miRNA to RNA-induced silencing complex (RISC) [142]. AGO2 silencing impairs miRNA pathways and downregulates RANK expression, reducing osteoclast differentiation and function [143]. Moreover, miR-100-5p could inhibit the occurrence and development of osteogenic bone metastasis in PCa via MTOR downregulation [137]. Therefore, miR-100-5p downregulation in PCa tissues can be related to osteoblastic and osteolytic lesions by enhancing osteoclast and osteoblast differentiation.

Wang et al. [144] found that miR-135b expression is downregulated in PCa tissues and that miR-135b upregulation inhibits signal transducer and activator of transcription 6 (STAT6) expression and reduces nuclear translocation of STAT6 for IL-4 in metastatic PCa cells. IL-4 is capable of inhibiting osteoclast differentiation through STAT6 nuclear translocation and activation, which inhibits RANKL-activated signaling and NF-κB activity, disturbing osteoclast activity and bone resorption [145,146]. It is suggested that miR-135b can modulate osteoclast activity in PCa cells by regulating STAT6 expression. Olivan et al. [147] suggested that miR-135b is involved in the bone homing of PC3-BM cells, affecting PCa bone metastasis via regulation of vesicle transport through interaction with t-SNAREs 1B (VIT1B), Janus kinase and microtubule interacting protein 2 (JAKMIP2), pleomorphic adenoma gene 1 (PLAG1), and platelet-derived growth factor subunit A (PDGFA), which are validated targets of miR-135b. Until now, VIT1b, JAKMIP2, and PLAG1 have not yet been described in PCa, but PDGFA is known to be associated with osteogenesis in PCa progression [148,149].

EGFR modulates PCa tumor growth, invasion, and bone metastasis [150] and seems to regulate several oncogenic genes, such as twist family BHLH transcription factor 1 (TWIST1) [151]. TWIST is overexpressed in PCa and promotes osteolytic bone lesions by modulating the expression of DKK1, an inhibitor of the Wnt pathway, and RUNX2, supporting osteogenesis induction and enhancing osteomimicry in prostate cells [152]. Chang et al. [151] indicated that EGFR modulates PCa bone metastasis by directly modulating miR-1 expression, which negatively regulates TWIST1 expression. Therefore, miR-1 downregulation enhances TWIST1 expression, leading to osteoblast mineralization and reduced stimulation of osteoclast differentiation.

miR-141-3p is downregulated in PCa cell lines (22RV1, PC-3, VCaP, DU145, and LNCaP) compared with a healthy prostate cell line (RWPE-1) and is known to act as a bone metastasis suppressor by inhibiting NF-κB activation via targeting of TRAF5 and TRAF6 [153], which are responsible for mediating NF-κB activation by RANK [106]. Therefore, miR-141-3p is capable of inhibiting osteoclast differentiation in metastatic PCa cells. By contrast, it was evidenced that miR-141-3p may support osteoblastic metastasis, and its expression is upregulated in the serum of patients with PCa bone metastasis [154] as well as in exosomes derived from MDA PCa 2b cells [155]. Higher levels of miR-141 in the serum were related to a higher number of bone lesions and serum alkaline phosphatase levels [154], which is considered an indicator of bone metastasis [156]. Furthermore, it was observed that MDA PCa 2b-derived exosomes enter osteoblasts and deliver miR-141-3p, promoting osteoblast activity via osteoprotegerin overexpression, an inhibitory cytokine of RANKL. miR-141-3p can upregulate osteoprotegerin expression by downregulating DLC1 Rho GTPase activating protein (DLC1), activating the p38/MAPK pathway and inducing osteoprotegerin/RANKL expression and osteoblast maturation [155]. Expression of miR-141-3p in PCa tissues remains controversial, but it is clear that this miRNA has an important role in bone metabolism and remodeling via modulation of osteoblast and osteoclast differentiation.

It is clear that miRNAs from PCa have a fundamental role in modulating the balance between osteoclast and osteoblast activity to stimulate tumor development and bone metastasis (Table 3). To date, miRNAs modulating osteoclastic differentiation are known to be the most suppressive miRNAs. Such regulation may be related to the fact that PCa primarily has an osteoblast phenotype. Here, miRNAs related to bone remodeling during PCa progression and metastasis were elucidated to better understand the role of miRNAs and to support future research on miRNAs as therapeutic targets.

## 5. miRNAs as Potential Therapeutic Targets

Since the United States Food and Drug Administration (FDA) approved small interfering RNA (siRNA) therapy for genetic conditions in 2018 [157,158], several investigations have been carried out by pharmaceutical companies using the noncoding RNA approach [159]. In posttranscriptional gene regulation, a strand of mature miRNA binds to AGO2, forming RISC, the same complex that is exploited by synthetic siRNAs to regulate gene expression, which is loaded on 3′- or 5′-untranslated regions (UTRs) of target mRNA.

siRNA and miRNA have some similarities, in that they are both short noncoding RNA molecules with 19–25 nucleotides (nt) exhibiting low in vivo stability, delivery challenges, and off-target effects [160]. However, whereas siRNAs need to bind with 100% complementarity to cleave the target sequence, miRNAs bind imperfectly to mRNA, with a minimum binding requirement of 2–8 nt [161]. Therefore, miRNAs may bind to multiple mRNA targets, being able to act dynamically as therapeutic agents. AntagomiRs and miRNA mimics are two important exogenous treatment tools based on miRNA technology. AntagomiRs are single-stranded RNA molecules with a degradation function that bind complementarily to a mature sequence of an overexpressed miRNA. By contrast, synthetic miRNAs mimic underexpressed targets with the objective of function restoration. In this case, consistent with siRNAs, it is suggested that double-stranded miRNAs are 100–1000-fold more potent, because of the need for maturation by the enzyme Dicer [162,163,164].

In 2013, Miravirsen, the first miRNA-targeted drug, was tested on 36 patients with chronic hepatitis C (HCV) infection in a phase 2 clinical trial. Miravirsen is a nucleic acid-modified DNA phosphorothioate antisense oligonucleotide that inhibits miR-122 function [165]. In the last miRBase actualization, there were over 2000 identified human miRNAs but not all their functions were available. Nevertheless, it is recognized that miRNAs may regulate up to 50% of genetic expression in humans [166], justifying the interest in the use of these molecules as antitumoral agents. A liposomal nanoformulation of miR-34a mimic was the first synthetic miRNA used for this purpose in humans, in a phase 1 study in adults with solid tumors refractory to standard treatments. Although the study was closed prematurely due to serious immune-mediated adverse effects, it demonstrated a dose-dependent modulation of relevant target genes and a proof-of-concept for miRNA-based cancer therapy [167]. Currently, 16 miRNAs are in the development phase for several human diseases, including miR-155 in phases 1 and 2 for lymphoma and leukemia conducted by MiRagen Therapeutics and miR-16 for mesothelioma by ENGeneIC [168,169,170,171].

miRNAs have not yet reached the clinical trial phase for PCa therapy; however, promising candidates have been identified, with bone metastasis appearing as one of the major research focuses. In 2009, Qin et al. [172] identified bone morphogenetic protein receptor 2 (BMPR2) as a target of miR-21 in PCa cells [172]. Inhibition of miR-154* and miR-379, DLK1-DIO3 cluster members, after cardiac inoculation of a PCa cell line in mice resulted in decreased bone metastasis and increased animal survival, stemming from downregulation of stromal antigen 2 (STAG2), a tumor suppressor gene [173]. Elevated expression of miR-409-3p/-5p was observed in bone metastatic PCa cell lines, and inhibitor-treated bone metastatic ARCaPM led to decreased bone metastasis and increased survival in mice [174]. Restoration of miR-133a-3p, -145, -204-5p, -582, and -466 inhibited bone metastasis in PCa in vivo through mediation of PI3K/AKT signaling, MYC/RAS regulation, and inactivation of NF-κB, TGF-β, and RUNX2, respectively [77,109,119,132,175]. Overexpression of miR-210-3p maintained the sustained activation of NF-κB signaling, resulting in EMT, invasion, migration, and bone metastasis of PC-3 cells [107]. By combining clinical samples with public databases, Liu et al. [176] found that miR-629-5p is increased in PCa metastasis, which leads to cell proliferation, migration, and invasion in vitro and promotes the growth of PCa cells in vivo by inhibiting A-kinase anchor protein 13 (AKAP13), a tumor suppressor. EVs derived from the murine PCa cell line RM1-BM enriched with miR-26a-5p, miR-27a-3p, and miR-30e-5p increased the metabolic activity, viability, and cell proliferation of osteoblast precursors, downregulated osteogenic markers, such as BMP2, and upregulated proinflammatory factors [177]. Moreover, exosomal miR-940 regulates osteogenic differentiation in PCa and is a potential therapeutic target in metastatic diseases [70].

Several miRNAs have been shown to induce significant changes in animal models of PCa apart from bone metastasis. AntagomiR delivery of miR-15a and miR-16 clusters in normal mouse prostate cells resulted in hyperplasia [178]. In CD44^+^ PCa stem cells, miR-34a is underexpressed; enforced expression inhibited metastasis and extended the survival of tumor-bearing mice [176]. One of the first-line drugs for metastatic PCa is docetaxel, an antimitotic chemotherapeutic. In docetaxel-resistant PCa cells, miR-34a, jointly with miR-27b, is downregulated, and sensibility returns with miRNA restoration [179]. Overexpression of the miR-17-92a cluster also increases docetaxel sensibility, as well as that of the antiandrogen drug Casodex, the AKT inhibitor MK-2206 2HCl [180]. Moreover, miR-30a is significantly downregulated in castration-resistant prostate cancer (CRPC) tissues, and overexpression of this miRNA reduces tumorigenicity in vivo by minimizing the expression of a distinct cell cycle protein, cyclin E2 (CCNE2) [181]. Lastly, the metastatic potential of PCa decreased with miR-200b overexpression in an orthotopic model, demonstrating the antiangiogenic activity of miRNA [182], given that neovascularization is a hallmark of cancer [183]. Currently, angiogenesis inhibitors are used concomitantly with chemotherapy in solid tumors, although none have been included in PCa treatment.

Other miRNAs deserve mention for their prostate antitumoral activity in vitro and *in vivo*, such as miR-338-5p/421, which abrogates serine peptidase inhibitor Kazal type 1 (SPINK1)-mediated oncogenesis [184], miR-1303, which regulates the Wnt/β-catenin pathway by targeting DKK3 [185], and miR-135a, which targets EGFR [186]. An miR-205-5p mimic nanoformulation was shown to have anticancer, antimetastatic, and chemosensitization potential together with docetaxel treatment [187].

It is also worth mentioning that natural plant products could modulate the expression of tumorigenic miRNAs. Metastasis-associated protein 1 (MTA1) is overexpressed in PCa, especially in bone metastatic lesions, and is related to decreased expression of E-cadherin through cathepsin B (CTSB), leading cells to EMT [188]. miR-22 was found to be positively associated with MTA1 and contribute to reducing E-cadherin expression [189]. Dietary polyphenols such as flavonoids and stilbenes, including resveratrol, are related to inhibition of miR-22 and the oncogenic MTA1-associated miRNAs miR-17 and miR-34a [190,191]. Pterostilbene, along with resveratrol, downregulates miR-17-5p and miR-106a-5p, miRNAs that target tumor suppressor phosphatase and tensin homolog (PTEN) [192,193]. In fact, animals supplemented with grape powder, with high content of resveratrol and stilbenes, presented lower circulating levels of oncogenic miR-34a and miR-22 and overall inflammation; thus, this plant product was suggested as a chemopreventive strategy in PCa progression [194].

Information about aberrant expressions of miRNAs in PCa and, more specifically, bone metastasis has increased, demonstrating that direct and indirect alterations made by mimics and antagomiRs may be an important tool in cancer therapy (Figure 2). In addition to antitumoral effects, the particularities of oligonucleotides, such as degradation by nucleases, low tissue permeability, and fast kidney excretion [195], must be taken into account during drug development.

## 6. Diagnosis

Early diagnosis of PCa bone metastasis may be one of the limiting factors of treatment success. Identification of molecular mechanisms associated with advances in radiology and nuclear medicine has led to the development of diagnostic imaging tools and biomarkers to improve PCa detection [196,197]. The most common primary clinical symptom of bone metastasis is pain [198,199]. According to the American Urological Association (AUA)/European Society for Medical Oncology (ESMO) guidelines, several routine analyses must be performed to investigate bone metastasis after confirming PCa, even if the patient does not report the presence of bone pain [200,201]. The standard-of-care imaging methods to detect bone metastasis include X-ray, bone scintigraphy, computed tomography, whole-body magnetic resonance imaging, and positron emission tomography–computed tomography [200,202].

Although imaging methods provide satisfactory results in the case of bone metastasis, alternative methods may achieve faster and more efficient detection. Alternative methods include bone biopsy [203] and identification of blood markers such as alkaline phosphatase [204,205] and calcium [206], which together with age-specific reference ranges, metastasis indicators, hemoglobin, Gleason score, treatment conditions, prostate-specific antigen (PSA), bone-specific alkaline phosphatase, corrected urinary N-telopeptide (uNTx), and absence of visceral metastasis, constitute the prognostic factors of PCa bone metastasis [207,208,209,210].

In addition to alternative methods, researchers are focusing on identifying the most accurate biomarkers that can be detected in body fluids by less invasive and safer procedures. In 2012, prostate cancer antigen 3 (PCA3), a long noncoding RNA, was the first urinary biomarker to gain FDA approval. It has higher sensitivity and specificity than serum PSA, helping to decide whether men with high PSA levels but negative prostate biopsy should undergo repeat biopsy [211]. Other promising urinary markers include fusion of the transmembrane protease serine 2 (TMPRSS2) and the ERG gene (TMPRSS2-ERG), homeobox C6 protein (HOXC6), and distal-less homeobox 1 (DLX1) [212,213,214]. PCA3 expression in the ejaculate, along with serine protease hepsin, is also proven to be a better predictor of PCa status and risk than serum PSA alone [215]. Seminal plasma is a robust source of markers such as prostatic acid phosphatase and alpha-methylacyl-CoA racemase (AMACR), given that 40% of semen is derived from prostatic tissue [216,217,218]. Although, to date, only PSA and PCA3 are approved by the FDA as biofluid markers, identification of the correct biomarker source is important for miRNA research, with urine, semen, serum, and plasma as the most cited and promising sources. These samples should be taken into consideration in study decisions [219].

Numerous papers have identified cancer-associated miRNA profiles exhibiting a highly abnormal expression that could be involved in metastasis development. Consequently, scientific efforts have been made to identify potential miRNA biomarkers and comprehend how they contribute to the development of PCa metastasis, aiming to accelerate the diagnostic process [210,220]. Porkka et al. [138], in 2007, identified a detailed PCa miRNA signature. On the basis of their findings, miRNAs were studied as biomarkers for PCa diagnosis and prognosis, including cases related to bone metastasis. Table 4 describes miRNAs with potential diagnostic and prognostic features for PCa bone metastasis, in addition to the miRNAs mentioned in this review. Many studies do not discriminate the metastatic site; thus, we discuss below some studies that identified miRNAs as possible biomarkers for diagnosis/prognosis of PCa bone metastasis.

miR-143 and miR-145 were some of the first miRNAs identified as possible biomarkers of PCa bone metastasis. In 2011, Peng et al. [221] described the downregulation of miR-143 and -145 in PCa patients with bone metastasis compared with those without bone metastasis. Additionally, miR-143 and -145 expression levels in primary PCa patients were negatively correlated with Gleason score and PSA level, suggesting that high miR-143 and -145 expressions might be associated with a lower risk of bone metastasis and better clinical state. Downregulation of miR-143 and -145 promotes migration, invasion, and EMT by reducing E-cadherin and increasing fibronectin expression in PC-3 cells [221]. In the following year, Huang et al. [222] observed a significant role of miR-143 and -145 in bone metastasis progression in PCa through regulation of cancer stem cell (CSC) properties [222].

Also in 2011, Sun et al. [223] identified miR-23b downregulation and miR-221 upregulation in most PCa bone metastasis tissues compared with normal prostate and primary PCa [223]. A few years later, Rice et al. [224] associated the downregulation of miR-23b with PC-3 invasion stemming from overexpression of Huntingtin-interacting protein 1-related (HIP1R) [224]. In 2018, Shao et al. [225] demonstrated that upregulation of miR-221-5p stimulates PC-3 cell proliferation, migration, and EMT via downregulation of SOCS1 and E-cadherin expression and activation of the RAS/RAF/MEK/ERK signaling pathway [225].

Zhiping et al. [226], in 2013, identified miR-181a upregulation in PCa metastasis in human tissues. The authors observed that, when upregulated in PC-3 cells in vitro, miR-181a stimulates EMT, invasion, and migration by suppressing TGF-β induced factor homeobox 2 (TGIF2) [226]. Additionally, miR-181a-5p upregulation, via suppression of migration and invasion inhibitory protein (MIIP), was found to inhibit Kruppel-like factor 17 (KLF17) and promote bone metastasis in vivo by activating EMT [227]. Recently, scientists have observed upregulation of this miRNA in EVs from the serum of PCa patients in metastatic bone groups compared with nonmetastatic bone groups [228].

In 2015, Zhang et al. [229] found that miR-188-5p was downregulated in bone metastatic patient samples compared with primary PCa samples, causing cell proliferation, invasion, and migration, being therefore associated with a poor prognosis. The authors observed a relationship between miRNA upregulation and suppression of lysosomal protein transmembrane 4 beta (LAPTM4B) [229]. Also in 2015, Fu et al. [230] reported the downregulation of miR-543 and -335 in PCa bone metastasis compared with the primary tumor in patient samples, associated with the overexpression of endothelial nitric oxide synthetase (eNOS). Upregulation of miR-335 and -543 suppressed eNOS expression and reduced PC-3 cell aggressiveness by decreasing cell migration and invasion [230].

mir-301a was investigated as a potential biomarker of PCa metastasis by Damodaran et al. [231] in 2016. The authors recorded miR-301a upregulation and consequent invasion and migration of PCa cells. The results revealed that the tumor and bone metastasis exhibited a significantly higher miR-301a expression (10-fold) than benign tissues [231]. In the same year, researchers discovered miR-320a downregulation in PCa patient samples, particularly in metastatic samples, including bone metastasis. Using PCa cell lines, such as PC-3, the authors observed increased cell migration and invasion associated with overexpression of lysosomal-associated membrane protein 1 (LAMP1), regulated negatively by miR-320a [232]. Additionally, Zhang et al. [233] demonstrated the downregulation of miR-194 in PC-3 cells, resulting in increased cell invasion via overexpression of MMP2 and MMP9 through the targeting of bone morphogenetic protein 1 (BMP1) [233]. Later, researchers found another target for miR-194, namely cadherin 2 (CDH2), confirming miRNA downregulation in PC-3 cells in vivo. Using an miR-194 mimic, they observed CDH2 downregulation and the consequent increase in cell death and apoptosis [234].

In 2019, Fan et al. [235] reported miR-127-3p downregulation in PCa tissues with bone metastasis. In vitro experiments revealed loss of migration and invasion with miRNA upregulation in PC-3 and C4-2B cells. Suppression of PCa bone metastasis in vivo was observed after miR-127-3p upregulation. A possible pathway was proposed: transcriptional downregulation of miR-127-3p by CCCTC-binding factor (CTCF) leads to overexpression of proteasome β-subunit 5 (PSMβ5), promoting bone metastasis in PCa [235]. miR-146a and miR-152 were investigated in the serum of 56 PCa and 56 healthy individuals from May 2009 to April 2017 at Jining First People’s Hospital, China. The researchers identified miR-146a upregulation and miR-152 downregulation in PCa patients *versus* healthy individuals and PCa bone metastasis *versus* PCa patients [236].

miR-199a-5p upregulation seems to decrease the expression of protein inhibitor of activated STAT3 (PIAS3), which increases AKT serine/threonine kinase 2 (AKT2) expression and enhances EMT to promote PCa metastasis, according to Tseng et al. [237]. Recently, another group showed that miR-199a-5p upregulation is associated with let-7a-5p downregulation, resulting in activation of the TGF-β pathway and promoting EMT, invasion, and migration in PC-3 cells and metastatic PCa cell lines [238].

Analysis of EVs from metastatic PCa cell lines revealed miR-425-5p upregulation in PC-3 cells compared with healthy prostate RWPE-1 and metastatic LNCaP cell lines. Bioinformatics evaluation identified suppression of the heat shock protein family B small member 8 (HSPB8), a target related to bone metastasis [239].

Recently, researchers reported miR-125a-3p, miR-330-3p, miR-339-5p, and miR-613 in EVs derived from the blood of PCa patients as potential biomarkers of bone metastasis. Only miR-125a-3p was downregulated in metastatic samples compared with nonmetastatic PCa; the other miRNAs were upregulated. The authors confirmed these findings in vitro using C4-2B cells. Bioinformatics analysis identified 25 targets of these four EV miRNAs related to bone metastasis [240]. Likewise, EVs derived from bone metastatic PCa were enriched with miR-378a-3p in patient serum, PC-3, and C4-2B cell lines, promoting osteolytic progression through activation of the DYRK1a/NFATC1/ANGPTL2 pathway [241].

**Table 4 cancers-15-04027-t004:** Potential miRNAs for the diagnosis/prognosis of bone metastasis in prostate cancer (PCa).

Expression	miRNA	Target	Bone Remodeling Signaling Pathways	Reference
Upregulated	miR-221	SOCS1	Stimulates cell proliferation, migration, and EMT via SOCS1 downregulation, E-cadherin expression, and activation of the RAS/RAF/MEK/ERK pathway.	[223,225]
	miR-181a	TGIF2 and KLF17	Stimulates EMT, invasion, and migration by suppressing TGIF2, inhibits KLF17 and promotes bone metastasis by activating EMT.	[226,228]
	miR-199a-5p	PIAS3 and TGF-β	Decreases PIAS3 expression, which increases AKT2 expression and enhances EMT to promote metastasis; activates TGF-β, promoting EMT, invasion, and migration.	[237,238]
	miR-425-5p	HSPB8	Suppresses HSPB8, increasing bone metastasis.	[239]
	miR-378-3p		Promotes osteolytic progression by activation of the DYRK1a/NFATC1/ANGPTL2 pathway.	[241]
Downregulated	miR-23b	HIP1R	Stimulates invasion due to HIP1R overexpression.	[223,224]
	miR-188-5p	LAPTM4B	Promotes cell proliferation, invasion, and migration by downregulating LAPTM4B.	[229]
	miR-543	eNOS	Increases eNOS expression, promoting cell migration and invasion.	[230]
	miR-335	eNOS	Increases eNOS expression, promoting cell migration and invasion.	[230]
	miR-320a	LAMP1	Increases cell migration and invasion via LAMP1 overexpression.	[232]
	miR-194	BMP1 and CDH2	Increases cell invasion via overexpression of MMP2 and MMP9 by targeting BMP1, decreases cell death and apoptosis via CDH2 overexpression.	[233,234]
	miR-127-3p	PSMB5	Promotes PSMB5 overexpression, increasing bone metastasis.	[235]
	let-7a-5p	TGF-β	Activates the TGF-β pathway, promoting EMT, invasion, and migration.	[238]

## 7. Conclusions and Future Directions

Many patients are diagnosed with PCa after presenting with bone pain, which is indicative of advanced stages of the disease. Skeletal commitment is frequent in PCa and related to pain, hyperkalemia, pathological bone fractures, metastatic epidural spinal cord compression, and cachexia, which may lead to death. Therefore, the development of well-defined diagnostic and treatment protocols for the local disease should be the major focus of research.

To the best of our knowledge, this was the first analysis of the relationship between miRNA expression profiles and bone metastatic PCa, substantiating the potential of these molecules to be included in new research protocols for disease diagnosis, prognosis, and treatment, following the global trend of clinical research on miRNAs.

## Figures and Tables

**Figure 1 cancers-15-04027-f001:**
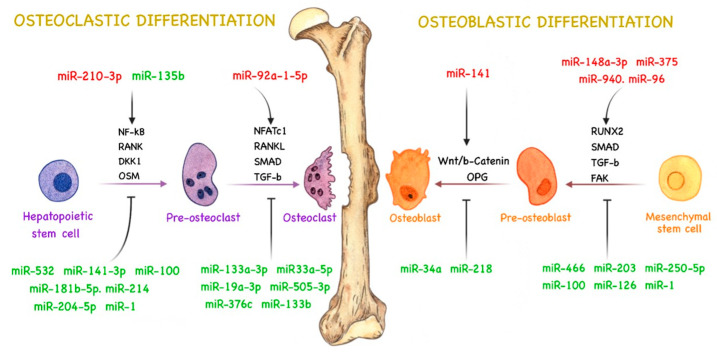
miRNAs and their role in prostate cancer (PCa) bone remodeling. Major activities of miRNAs in bone remodeling during PCa progression. miRNAs can directly or indirectly affect different molecular mechanisms related to osteoblast and osteoclast differentiation to promote tumor development and bone metastasis. miRNAs highlighted in red are upregulated in PCa, whereas those highlighted in green are downregulated. Blunt arrows (┳) indicate pathway inhibition, and sharp arrows (→) indicate pathway stimulation.

**Figure 2 cancers-15-04027-f002:**
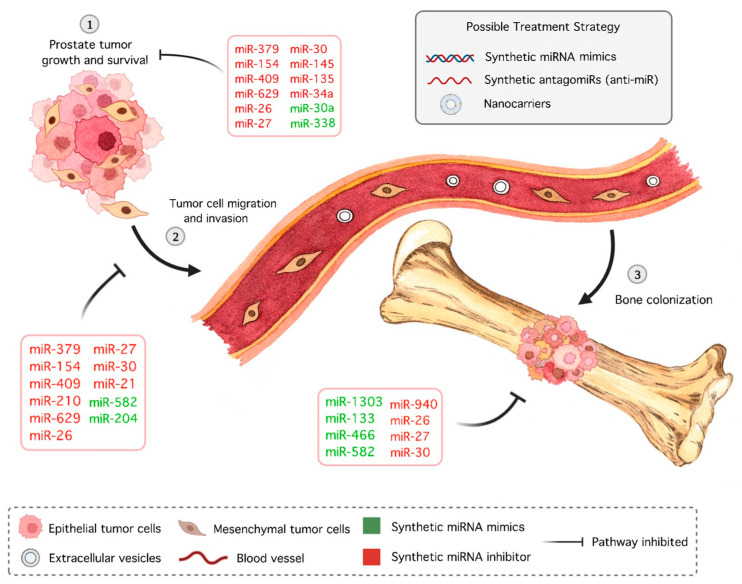
miRNA-based therapeutic strategies and their inhibitory effects on prostate cancer bone metastasis. Schematic representation of prostate cancer bone metastasis and the expected effects of therapeutic strategies based on synthetic miRNA mimics or inhibitor nanoformulations that minimize molecule degradation. Synthetic miRNAs potentially act by ① modulating prostate tumor maintenance, growth, and/or survival; ② regulating epithelial–mesenchymal transition and reducing the ability of tumor cells to migrate and invade surrounding tissues; and ③ modulating bone remodeling by regulating interactions between tumor cells and the bone microenvironment during bone colonization.

**Table 1 cancers-15-04027-t001:** miRNAs related to osteoblast activity in prostate cancer (PCa).

Expression	miRNA	Target	Bone Remodeling Signaling Pathways	Reference
Upregulated	miR-96	AKT1S1	Inhibits AKT1S1 expression, inducing MTOR kinase activity and osteoblast differentiation.	[83]
	miR-96	E-cadherin and EPCAM	Enhances PCa cell–cell interactions and their ability to bind to osteoblasts by upregulating E-cadherin and EPCAM expression.	[85]
	miR-940	ARHGAP1 and FAM134A	Promotes differentiation of mesenchymal stem cells to osteoblasts.	[70]
	miR-148a-3p	-	Induces osteogenic differentiation.	[101]
	miR-375	-	Enhances osteoprotegerin, RUNX2, osteopontin, and bone sialoprotein expression in LNCaP cells, stimulating osteoblast differentiation and function.	[102,103]
Downregulated	miR-446	RUNX2	Suppresses PCa proliferation and bone metastasis through regulation of osteogenic factors, such as RUNX2, osteopontin, osteocalcin, ANGPT1, ANGPT4, MMP11, Fyn, pAKT, FAK, and vimentin.	[77]
	miR-203	RUNX2	Negatively regulates RUNX2 expression, suppressing bone formation and metastasis.	[78]
	miR-126	VCAM-1	Osteoblast-derived WISP-1 induces miR-126 downregulation via αvβ1 integrin, FAK, and p38 signaling pathways, leading to migration and VCAM-1 expression in metastatic PCa cells.	[81]
	miR-218	LGR4	Suppresses IL-6-induced cell proliferation and invasion via downregulation of LGR4.	[88]
	miR-34a	TCL7	Inhibits TCF7 expression, downregulating the Wnt/β-catenin pathway.	[94]
	miR-205-5p	RUNX2	Negatively regulates RUNX2 expression, suppressing osteogenic differentiation in hBMSC.	[95,96]

**Table 2 cancers-15-04027-t002:** miRNAs related to osteoclast activity in prostate cancer (PCa).

Expression	miRNA	Target	Bone Remodeling Signaling Pathways	Reference
Upregulated	miR-92a-1-5p	COL1A1	Induces type I collagen degradation by targeting COL1A1, stimulating bone ECM degradation and bone resorption.	[101]
	miR-210-3p	TNIP1 and SOCS1	Promotes osteoclast differentiation by sustained activation of the NF-κB signaling pathway through TNIP1 and SOCS1 inhibition.	[107]
Downregulated	miR-532-3p	TRAF1, TRAF2, and TRAF3	Suppresses NF-κB activation via downregulation of TRAF1, TRAF2, and TRAF3.	[108]
	miR-204-5p	TRAF1, TAB3, and MAP3K3	Suppresses NF-κB activation via downregulation of TRAF1, TAB3, and MAP3K3.	[109]
	miR-214		Promotes NF-κB activity, leading to osteoclast differentiation.	[112]
	miR-376c	MMP2	Negatively regulates MMP2 expression, suppressing matrix degradation and osteoclastogenesis.	[113]
	miR-181b-5p	OSM	Inhibits OSM expression, decreases IL-6 and AREG, and increases osteoprotegerin, suppressing osteoclast differentiation.	[118]
	miR-133a-3p	EGFR, FGFR1, IGF1R, and MET	Directly inhibits cytokine receptors of the PI3K/AKT signaling pathway, minimizing stimulation of osteolytic bone lesions.	[119]
	miR-582-3p and miR-582-5p	SMAD2, SMAD4, TGFβRI, and TGFβRII	Inhibits TGF-β signaling activity by downregulating SMAD2, SMAD4, TGFβRI, and TGFβRII, reducing bone osteolytic metastasis.	[132]
	miR-505-3p	SMAD2 and SMAD3	Inhibits TGF-β signaling activity via downregulation of SMAD2 and SMAD3, reducing invasion and bone metastasis.	[133]
	miR-19a-3p	SMAD2 and SMAD4	Inhibits TGF-β signaling activity via downregulation of SMAD2 and SMAD4, reducing osteolytic bone lesions.	[134]
	miR-133b	TGFβRI and TGRFβRII	Inhibits TGF-β signaling activity via downregulation of TGFβRI and TGRFβRII, reducing osteolytic bone lesions.	[135]
	miR-33a-5p	TGFβRI	Inhibits TGF-β signaling activity via downregulation of TGFβRI, reducing osteolytic bone lesions.	[49]

**Table 3 cancers-15-04027-t003:** miRNAs related to osteoclast and osteoblast activities in prostate cancer (PCa).

Expression	miRNA	Target	Bone Remodeling Signaling Pathways	Reference
Upregulated	141-3p	DLC1	Activates the p38MAPK pathway, increasing osteoprotegerin/RANKL expression and osteoblast maturation via downregulation of DLC1.	[155]
Downregulated	141-3p	TRAF5 and TRAF6	Suppresses NF-κB activation via downregulation of TRAF5 and TRAF6.	[153]
	miR-100	AGO2	Suppresses osteoclast differentiation and function by impairing miRNA pathways through AGO2 inhibition.	[141,143]
	miR-100-5p	MTOR	Inhibits osteoblast differentiation and function via MTOR downregulation.	[137]
	miR-135b	STAT6	Promotes osteoclast activity and bone resorption by stimulating RANKL-activated signaling and NF-κB activity via downregulation of STAT6.	[144]
	miR-135b	VIT1b, JAKMIP2, PLAG1, and PDGFA	Implicates osteogenesis in PCa via regulation of VIT1b, JAKMIP2, PLAG1, and PDGFA genes.	[147]
	miR-1	TWIST1	Regulates TWSIT1 expression, which promotes PCa bone remodeling by regulating DKK1 and RUNX2 expression.	[151]

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
