# Peer review of "The Bone Microenvironment Soil in Prostate Cancer Metastasis: An miRNA Approach"

_cancers, 2023, doi:10.3390/cancers15164027_

Round 1

Reviewer 1 Report

Prostate cancer (PCa) is men's second most prevalent malignancy worldwide. While PCa has excellent cure rates when treated locally, it has a high risk of death when it is progressed. Additionally, bone metastasis PCa has been linked to significant mortality risk. Exosome trafficking, which also establishes the potential metastatic niche in bone, is expected to facilitate the deregulation of microRNA (miRNA) production, which has been linked to tumor growth. In this review, the authors summarize the important role of miRNAs in developing, diagnosing, and treating bone metastasis. The article is well organized and easy to understand for the readers. Tables are the main strength of this article. Hence, I recommend acceptance.

Author Response

We would like to thank the editor and reviewers for reviewing our work. In the revised version, we have included two paragraphs as suggested (pages 13 and 14; lines 540-552, dealing with natural products, and page 15, lines 588 to 603, dealing with different biofluids). The manuscript has been grammatically revised. Figure 2 was also corrected.
Important changes are highlighted.
Yours sincerely
Tania

Reviewer 2 Report

This manuscript that submitted by Prigol et al. summarized the role of miRNA on the bone metastasis of prostate cancer. There are rare review papers which summarized the significant findings on bone metastasis. This manuscript is good writing and good organized. Just one suggestion for the authors, it would be beneficial for audiences that authors could provide a paragraph to summarize whether there had established the biofluid analysis for progression of prostate cancer. 

The scientific writing is good and easy to read.

Author Response

(The authors gave the same response as above.)
